# Enhancement of Vaccine-Induced T-Cell Responses by PD-L1 Blockade in Calves

**DOI:** 10.3390/vaccines11030559

**Published:** 2023-03-01

**Authors:** Tomohiro Okagawa, Satoru Konnai, Hayato Nakamura, Otgontuya Ganbaatar, Yamato Sajiki, Kei Watari, Haruka Noda, Mitsuru Honma, Yukinari Kato, Yasuhiko Suzuki, Naoya Maekawa, Shiro Murata, Kazuhiko Ohashi

**Affiliations:** 1Department of Advanced Pharmaceutics, Faculty of Veterinary Medicine, Hokkaido University, Sapporo 060-0818, Japan; 2Department of Disease Control, Faculty of Veterinary Medicine, Hokkaido University, Sapporo 060-0818, Japan; 3Institute for Vaccine Research and Development (HU-IVReD), Hokkaido University, Sapporo 001-0021, Japan; 4Hokkaido Research Station, Snow Brand Seed Co., Ltd., Naganuma 069-1464, Japan; 5Department of Antibody Drug Development, Tohoku University Graduate School of Medicine, Sendai 980-8575, Japan; 6Department of Molecular Pharmacology, Tohoku University Graduate School of Medicine, Sendai 980-8575, Japan; 7Division of Bioresources, International Institute for Zoonosis Control, Hokkaido University, Sapporo 001-0020, Japan; 8Global Station for Zoonosis Control, Global Institution for Collaborative Research and Education (GI-CoRE), Hokkaido University, Sapporo 001-0020, Japan; 9International Affairs Office, Faculty of Veterinary Medicine, Hokkaido University, Sapporo 060-0818, Japan

**Keywords:** PD-L1, PD-1, T cell, live attenuated vaccine, immunotherapy, cattle

## Abstract

Interactions between programmed death 1 (PD-1) and PD-ligand 1 (PD-L1) cause functional exhaustion of T cells by inducing inhibitory signals, thereby attenuating effector functions of T cells. We have developed an anti-bovine PD-L1 blocking antibody (Ab) and have demonstrated that blockade of the interaction between PD-1 and PD-L1 reactivates T-cell responses in cattle. In the present study, we examined the potential utility of PD-1/PD-L1-targeted immunotherapy in enhancing T-cell responses to vaccination. Calves were inoculated with a hexavalent live-attenuated viral vaccine against bovine respiratory infections in combination with treatment with an anti-PD-L1 Ab. The expression kinetics of PD-1 in T cells and T-cell responses to viral antigens were measured before and after vaccination to evaluate the adjuvant effect of anti-PD-L1 Ab. PD-1 expression was upregulated in vaccinated calves after the administration of a booster vaccination. The activation status of CD4^+^, CD8^+^, and γδTCR^+^ T cells was enhanced by the combination of vaccination and PD-L1 blockade. In addition, IFN-γ responses to viral antigens were increased following combinatorial vaccination with PD-L1 blockade. In conclusion, the blockade of the PD-1/PD-L1 interaction enhances T-cell responses induced by vaccination in cattle, indicating the potential utility of anti-PD-L1 Ab in improving the efficacy of current vaccination programs.

## 1. Introduction

Vaccines are central to the control strategy for infectious diseases by preventing illness and death in individuals and the spread of disease through populations. In veterinary medicine, a variety of vaccines have been developed and used for the control of infectious diseases in animals. The prevention of infection by vaccination is particularly important for livestock raised in groups. In calves, whose immune systems are immature and naive to various pathogens, immunization with vaccines is essential and effective vaccine programs are required to protect calves against infectious diseases. Various adjuvants are currently added to vaccine products to enhance their efficacy. Live attenuated vaccines are the most widely used vaccines for livestock and can induce both cellular and humoral immunity [1].

Programmed death 1 (PD-1) is an immune checkpoint molecule predominantly expressed by T cells [2]. The binding of PD-1 to its ligand programmed death-ligand 1 (PD-L1), which is expressed by antigen-presenting cells and immune cells, inhibits T-cell receptor (TCR) signaling and reduces the effector function of T cells in a process known as T-cell exhaustion [3,4]. Previous studies in cattle have reported that PD-1 expression is upregulated in T cells and PD-L1 expression is upregulated in antigen-presenting cells and B cells during chronic infections, thereby decreasing T-cell responsiveness to antigens and Th1 cytokine responses [5,6,7,8]. Conversely, the effector function of T cells can be reactivated by inhibiting PD-1/PD-L1 binding using anti-PD-L1 antibody (Ab) [9,10]. We previously developed an anti-bovine PD-L1 rat-bovine chimeric Ab (chAb) which is able to block bovine PD-1/PD-L1 binding [10]. In clinical trials using cattle, treatment with anti-bovine PD-L1 chAb activated T-cell responses, including cell proliferation and Th1 cytokine production [10,11,12]. Thus, anti-bovine PD-L1 chAb is expected to have potential as a molecular-targeted drug for inducing immune responses by preventing T-cell exhaustion in cattle.

T cells play an important role in the induction of immune memory following vaccination with attenuated live viruses [1]. However, the PD-1/PD-L1 pathway is posited to inhibit the induction of memory T cells responsible for cellular immunity in response to vaccines or infection [13]. A previous study in a mouse model of lymphocytic choriomeningitis virus (LCMV) infection reported that the blockade of PD-1/PD-L1 inhibitory signals in exhausted CD8^+^ T cells in combination with therapeutic vaccination against LCMV synergistically enhanced the functional responses of CD8^+^ T cells and decreased viral titers in mice chronically infected with LCMV [14]. Furthermore, PD-1 inhibition by anti-PD-1 Ab or soluble PD-1 protein has been shown to enhance vaccine-induced primary and memory CD8^+^ T-cell responses, thereby producing an adjuvant effect via a novel mechanism of action, in mouse and rhesus macaque models [15,16]. However, studies on the adjuvant effect of PD-1/PD-L1 inhibition have been limited to mouse and monkey experimental models, with no studies to date examining the combinatorial efficacy of anti-PD-L1 or anti-PD-1 Abs with vaccines against viral infection in clinical settings.

Accordingly, the present study aimed to evaluate the adjuvant effect of a PD-1/PD-L1 inhibitor as an adjuvant to vaccination in cattle using a currently used commercial vaccine and anti-bovine PD-L1 chAb. Two doses of a hexavalent vaccine against viruses causing bovine respiratory infections were administered to calves in combination with anti-PD-L1 chAb. Changes in PD-1 expression by T cells, numbers of viral antigen-responding T cells, and cytokine response were measured before and after vaccination.

## 2. Materials and Methods

### 2.1. Animals, Vaccination, and Sample Collection

The design of the animal experiment using calves is shown in Figure 1. Twelve male calves of the Holstein breed ranging from two to three months of age were used in the present study. All calves received two intramuscular administrations of “KYOTOBIKEN” Calfwin 6 Combo Live Vaccine (Kyoto Biken Laboratories, Uji, Japan), a hexavalent live-attenuated vaccine which consists of bovine respiratory syncytial virus (BRSV, rs-52 strain, >10^5.0^ TCID_50_), bovine viral diarrhea virus (BVDV) type 1a (No.1255 strain, >10^3.0^ TCID_50_), BVDV-2 (KZ1254 strain, >10^3.0^ TCID_50_), bovine herpesvirus-1 (BHV-1, No.758-43 strain, >10^4.0^ TCID_50_), parainfluenza virus type 3 (PIV3, BN-CE strain, >10^5.0^ TCID_50_), and adenovirus type 7 (TS-GT strain, >10^3.0^ TCID_50_) with a six-week interval. Six of the vaccinated animals were intravenously administered 2 mg/kg of anti-bovine PD-L1 rat-bovine chAb (Boch4G12) [10] twice just after the primary and booster vaccinations. Peripheral blood samples were collected from all calves at least once every two weeks following the primary vaccination. Peripheral blood mononuclear cells (PBMCs) were isolated from blood samples by density gradient centrifugation using Percoll (GE Healthcare, Chicago, IL, USA), washed three times with phosphate-buffered saline (PBS), and suspended in PBS. Fresh PBMCs or PBMCs cryopreserved in heat-inactivated fetal bovine serum (FBS; Thermo Fisher Scientific, Waltham, MA, USA) containing 10% dimethyl sulfoxide (Nacalai Tesque, Kyoto, Japan) were used in subsequent experiments. All experimental procedures were conducted with the approval of the Ethics Committee of the Faculty of Veterinary Medicine, Hokkaido University (approval #17-0024).

### 2.2. Flow Cytometric Analysis of PD-1 Expression

Six-color analysis of PD-1-expressing T cells was performed using cryopreserved PBMCs obtained from animals before and after booster vaccination. PBMCs were thawed and incubated in PBS containing 10% goat serum (Thermo Fisher Scientific) at 25 °C for 15 min to prevent nonspecific reactions. Cells were then washed and incubated with anti-bovine PD-1 monoclonal antibody (1D10F1, rabbit IgG, in-house) or rabbit IgG isotype control (EPR25A, Abcam, Cambridge, UK) in the presence of anti-TCR1-N24 mAb (GB21A, mouse IgG_2b_; Washington State University Monoclonal Antibody Center, Pullman, WA, USA) at 25 °C for 30 min. After washing with PBS containing 1% bovine serum albumin (BSA; Sigma–Aldrich, St. Louis, MO, USA), cells were stained with Alexa Fluor 647-conjugated anti-rabbit IgG(H+L) goat F(ab’)2 (Thermo Fisher Scientific), FITC-conjugated anti-mouse IgG_2b_ goat F(ab’)2 (Beckman Coulter, Fullerton, CA, USA), PerCp/Cy5.5-conjugated anti-bovine CD3 mAb (MM1A; Washington State University Monoclonal Antibody Center), PE/Cy7-conjugated anti-bovine CD4 mAb (CC8; Bio-Rad, Hercules, CA, USA), PE-conjugated anti-bovine CD8 mAb (CC63, Bio-Rad), and Fixable Viability Dye eFluor 780 (Thermo Fisher Scientific) at 25 °C for 30 min. MM1A and CC8 mAbs were conjugated with PerCp/Cy5.5 and PE/Cy7, respectively, using Lightning-Link Conjugation Kits (Abcam). Cells were then washed and immediately analyzed using FACSLyric (BD Biosciences) and FACSuite software (BD Biosciences).

### 2.3. PBMC Cultivation Assay

T-cell responses to vaccination were analyzed using a cultivation assay of PBMCs. Fresh PBMCs (1 × 10^6^) were cultured with 5 × 10^6.0^ TCID_50_/mL of UV-inactivated BRSV (NMK7 strain), 5 × 10^6.0^ TCID_50_/mL of UV-inactivated BVDV-1a (No.12 strain), or 5 μg/mL of concanavalin A (Con A; Sigma–Aldrich) in triplicate in 96-well round-bottom microplates (Corning Inc., Corning, NY, USA) containing 100 μL of RPMI1640 Medium (Sigma–Aldrich) supplemented with 10% heat-inactivated FBS (Thermo Fisher Scientific), 100 IU/mL of penicillin, 100 μg/mL of streptomycin, and 0.01% L-glutamine (Thermo Fisher Scientific) at 37 °C with 5% CO_2_ for seven days. After cultivation, cells were harvested and used for flow cytometric analysis. Culture supernatants were collected and stored at −30 °C. IFN-γ concentrations in supernatants were measured using ELISA (Bovine IFN-γ ELISA Basic Kit; Mabtech, Nacka Strand, Sweden) in duplicate according to the manufacturer’s protocol.

Cultured PBMCs were collected and incubated in PBS containing 10% goat serum (Thermo Fisher Scientific) as described above. Cells were then washed and labeled with Alexa Fluor 488-conjugated anti-bovine CD25 mAb (CACT116A; VMRD, Pullman, WA, USA), Alexa Fluor 647-conjugated anti-bovine CD69 mAb (KTSN7A; Washington State University Monoclonal Antibody Center), PerCp/Cy5.5-conjugated anti-bovine CD3 mAb (MM1A; Washington State University Monoclonal Antibody Center), PE/Cy7-conjugated anti-bovine CD4 mAb (CC8; Bio-Rad), PE-conjugated anti-bovine CD8 mAb (CC63, Bio-Rad), and APC/Cy7-conjugated anti-TCR1-N24 mAb (GB21A; Washington State University Monoclonal Antibody Center) for 20 min at 4 °C. MM1A, CC8, and GB21A mAbs were conjugated with corresponding fluorochromes using Lightning-Link Conjugation Kits (Abcam). CACT116A and KTSN7A mAbs were prelabeled with Zenon Mouse IgG_1_ Labeling Kits (Thermo Fisher Scientific). Cells were then washed with 1% BSA-PBS and immediately analyzed using FACSVerse (BD Biosciences) and FACSuite software (BD Biosciences).

### 2.4. Statistical Analyses

All statistical tests were performed in GraphPad Prism 6 (GraphPad Software, San Diego, CA, USA). *p*-values less than 0.05 were considered statistically significant.

## 3. Results

### 3.1. Kinetics of PD-1 Expression by T Cells in Vaccinated Calves

PD-1 expression on T-cell subsets was analyzed by flow cytometry of PBMCs isolated from calves before and after vaccinations with Calfwin 6 with or without the administration of anti-PD-L1 chAb (Appendix A). There was a trend toward an increased proportion of PD-1^+^ cells in each T-cell subset after primary and booster vaccinations (Appendix A). In particular, PD-1 expression was upregulated in CD3^+^CD4^+^, CD3^+^CD8^+^, and CD3^+^γδTCR^+^ T cells at day 49 after booster vaccination in calves that received the vaccine only (V group), compared with before the booster vaccination (at day 42; Figure 2A–C). In contrast, no upregulation of PD-1 expression was observed in any T-cell subset in vaccinated calves treated with anti-PD-L1 Ab (VA group) at day 49 (Figure 2A–C). Thus, two-dose vaccination with Calfwin 6 induced a higher proportion of PD-1^+^ exhausted T cells in calves. These findings indicate that the PD-1/PD-L1 pathway may inhibit the induction of T-cell responses following vaccination in calves.

### 3.2. Activation of T Cells by the Combination of Vaccination and PD-L1 Blockade in Calves

To evaluate the adjuvant effect of the combinatorial use of anti-bovine PD-L1 antibody and vaccination, T-cell responses to BRSV and BVDV-1 antigens were analyzed. First, PBMCs were stimulated with BRSV and BVDV-1 antigens and the expression levels of CD25 and CD69, as activation-induced markers (AIM) [17], were analyzed in CD3^+^CD4^+^, CD3^+^CD8^+^, or CD3^+^γδTCR^+^ T cells by flow cytometry (Appendix A). After booster vaccination (at day 70), CD3^+^CD4^+^, CD3^+^CD8^+^, and CD3^+^γδTCR^+^ T-cell responses to BRSV or BVDV-1 antigen were observed, with antigen stimulation increasing the population of CD25^+^CD69^+^ cells in each T-cell subset of cultivated PBMCs (Appendix A).

The frequency of CD25^+^CD69^+^ cells was significantly increased in CD3^+^CD4^+^ T cells stimulated with BRSV and BVDV-1 in the VA group after the booster vaccination at day 70 (Figure 3A). In addition, the frequency of AIM^+^ cells (CD25^+^CD69^+^, CD25^+^CD69^−^, and CD25^−^CD69^+^ cells) in the CD3^+^CD4^+^ T-cell population was increased in stimulated PBMCs in the VA group at day 70 (Figure 3B). The AIM^+^ responses of CD4^+^ T cells to BVDV-1 antigen were also significantly enhanced at day 28 after primary vaccination (Figure 3B). The kinetics of CD4^+^ T-cell activation demonstrated strong induction after the booster vaccination in the VA group. On the other hand, CD4^+^ T-cell activation was not strongly induced even after the booster vaccination, except in V5 (Appendix A). Thus, CD4^+^ T-cell responses to the vaccine were enhanced when the vaccine was combined with the administration of anti-PD-L1 Ab in calves.

We then evaluated the activation of the other T-cell subsets, including CD8^+^ and γδTCR^+^ T cells, before and after vaccination. In the CD3^+^CD8^+^ T-cell population of BRSV- and BVDV-1-stimulated PBMCs, the proportion of CD25^+^CD69^+^ cells was significantly increased in the VA group after primary vaccination at day 28 (Figure 4A). In the V group, the frequency of CD25^+^CD69^+^CD8^+^ cells was increased following BVDV-1 stimulation but not BRSV stimulation (Figure 4A). Similar patterns were observed for the proportion of AIM^+^CD8^+^ T cells (Figure 4B). In particular, the frequency of BRSV-reactive AIM^+^CD8^+^ T cells was significantly higher in the VA group compared with the V group at day 28 (Figure 4B). These results indicate that more than one administration of the vaccine combined with PD-L1 blockade can also induce CD8^+^ T-cell responses to the vaccine.

In calves, γδ T cells are the most abundant subset of T cells in blood [18]. γδ T cells have also been reported to be involved in vaccine responses as effector cells in cattle [19,20,21], making the analysis of γδ T cells essential for immunological evaluations of vaccine programs in calves. AIM analysis by flow cytometry demonstrated that the frequencies of CD25^+^CD69^+^ cells and AIM^+^ cells in the CD3^+^γδTCR^+^ T-cell population were significantly increased in PBMCs stimulated with BRSV and BVDV-1 in the VA group at day 70 compared with those at day 14 (Figure 5A,B). In the V group, fractions in the CD3^+^γδTCR^+^ T-cell population stimulated by BVDV-1 were significantly increased at day 70 but not in the cells stimulated with BRSV (Figure 5A, B). Although the antigen specificity of γδ T cells during vaccination remains unclear, we found that γδ T cells are also involved in the immune response to this vaccination, and that this response was enhanced by PD-L1 blockade. Taken together, AIM assays using flow cytometry demonstrated that the combination of vaccination and PD-L1 inhibition enhanced the activation of vaccine-responsive T cells.

### 3.3. Enhancement of IFN-γ Response by the Combination of Vaccination and PD-L1 Blockade in Calves

Th1 cytokine responses to the vaccine were examined as a functional parameter of T cells. In this experiment, IFN-γ responses induced by antigen stimulation were measured as a representation of Th1 cytokine responses. PBMCs from calves vaccinated or unvaccinated with Calfwin 6 were cultured in the presence of BRSV or BVDV-1 antigen and anti-PD-L1 Ab, with IFN-γ production measured by ELISA. There was a trend toward increased IFN-γ responses to antigens following PD-L1 blockade in vaccinated animals only (Appendix A). This finding indicates that this vaccine in calves increased the proportion of functionally exhausted T-cell populations via activation of the PD-1/PD-L1 pathway.

Comparing IFN-γ responses in calves treated with anti-PD-L1 Ab before vaccination and after the booster vaccination, there was a trend toward increased IFN-γ responses to BRSV in the VA group at day 70 (Figure 6A). The mean concentration of IFN-γ was 0.02 and 1.05 ng/mL in BRSV-stimulated PBMCs in the V and VA groups, respectively (Figure 6A). Furthermore, the IFN-γ response to BVDV-1 stimulation was significantly higher in the VA group at day 70 (mean, 8.16 ng/mL) compared with day 14 (mean, 1.71 ng/mL; Figure 6B), indicating that a Th1 cytokine response to the antigen was induced by the combinatorial use of vaccination and PD-1/PD-L1 blockade. In summary, these results demonstrate that combinatorial treatment with vaccination and anti-PD-L1 Ab functionally activates T cells and enhances Th1 cytokine responses to antigen in vaccinated calves.

## 4. Discussion

Calf pneumonia has been the greatest cause of economic loss in the cattle industry [22,23]. Viruses such as BRSV, BVDV, IBRV, and PI3V cause primary infection in calves leading to the accumulation of environmental stress [24]. Secondary and tertiary bacterial infections may then develop, resulting in severe mixed infections that lead to death or lifelong growth failure in infected calves [24]. As a first line of defense against respiratory infections in calves, multivalent vaccines against viral pathogens have been developed and incorporated into calf vaccination programs [25,26,27]. However, some calves still develop infections despite repeated vaccinations, indicating insufficient induction of adaptive immune responses. In the development and evaluation of vaccines, the induction of neutralizing antibodies is the most important factor, and the analysis of T-cell responses is often neglected [27,28,29,30,31]. Accordingly, there are limited data regarding the induction of T-cell responses following the administration of commercially available vaccines.

Immunotherapy targeting the PD-1/PD-L1 pathway has been widely applied in the clinical treatment of human tumors. PD-1/PD-L1 blockade has also been studied and developed in veterinary medicine as a novel treatment for canine tumors and infectious diseases in domestic animals [11,12,13,32,33,34,35,36]. In addition, clinical trials of the combinatorial use of therapeutic vaccines against tumors and PD-1/PD-L1 inhibitors are currently underway in humans [37]. However, there are currently no reported studies combining vaccination and PD-1/PD-L1 blockade in veterinary medicine. In addition, there are limited reported studies of the combined use of live-attenuated vaccines against viruses and PD-1/PD-L1 inhibitors. In the present study, we addressed these issues by validating the immunological effect of the combination of vaccination and anti-PD-L1 Ab in a bovine vaccine model.

We observed an increased proportion of PD-1^+^ cells in all T-cell subsets after vaccination in calves. This upregulation was more prominent after booster vaccination in the V group but not in the VA group, indicating T-cell activation and cytokine responses were enhanced after booster vaccination in the VA group. In addition, slight differences in responses to PD-L1 blockade were observed between CD4^+^, CD8^+^, and γδTCR^+^ T-cell subsets, including increases in the rate and potency of T-cell activation. Although PD-1 expression levels and responsiveness to PD-L1 blockade may be correlated in T-cell fractions, no clear difference was observed due to the limited number of animals included in the present study (data not shown). The specific T-cell fractions, such as memory or effector T-cell subpopulations, responsible for mediating the effects of PD-L1 inhibition remain unclear. To determine the mechanism underlying the induction of T-cell responses following the vaccination with attenuated viruses, phenotypic and transcriptomic analyses of T cells following combinatorial use of vaccination and PD-L1 blockade are warranted.

In the present study, we evaluated T-cell responses using AIM assays and cytokine measurements. The AIM assay is a technique used to evaluate vaccination efficacy in humans by measuring multiple markers of T-cell responses [17,38]. Furthermore, numbers of AIM^+^ T cells is known to be correlated with numbers of cytokine-producing T cells in response to vaccine antigens [17]. In the present study, IFN-γ was measured as a representative cytokine produced by effector T cells. The results of the AIM and IFN-γ assays were consistent. Although the AIM assay is not widely used in bovine immunology, this technique may have utility in further studies of T-cell responses to vaccination in cattle.

T-cell responses to two viral antigens in the hexavalent vaccine, BRSV and BVDV-1, were examined in the present study and found to be inhibited by the PD-1/PD-L1 pathway after the vaccination, although some differences were observed in the responses produced by PD-L1 inhibition. The induction of T-cell responses in addition to humoral responses is considered important for complete protective immunity against BRSV and BVDV [25,26]. In RSV infection, IFN-γ is a key cytokine that mediates the antiviral Th1 responses that activate natural killer cells and cytotoxic CD8^+^ T cells, thereby resulting in viral clearance and recovery from a viral challenge in calf and mouse models [39,40,41,42,43]. On the other hand, BVDV-specific CD4^+^ and CD8^+^ T cells play essential roles in protecting against BVDV infection cattle in the absence of a BVDV-neutralizing Ab response [44,45,46,47]. However, the present study did not examine the protective effects of the combinatorial use of vaccination and PD-L1 blockade against viral challenge. Viral challenge experiments should be conducted to further validate the efficacy of PD-L1 blockade as an adjuvant to vaccination in calves.

In conclusion, the results of the present study demonstrate the potential utility of PD-1/PD-L1 inhibitors as vaccine adjuvants in cattle. The combination of viral vaccination and PD-L1 blockade may enhance vaccine efficacy and reduce the number of calf deaths due to pneumonia and diarrhea, thereby improving dairy and beef cattle productivity. In addition, T-cell exhaustion via the PD-1/PD-L1 pathway is a conserved immune regulatory mechanism from domestic and companion animals to humans [2,5,6,48,49,50,51,52,53,54]. Accordingly, this strategy may increase the efficacy of vaccination programs across a wide range of animal species, not just in cattle. Further studies using other animal species and vaccines are required to determine the potential efficacy of PD-1/PD-L1 inhibitors as vaccine adjuvants.

## Figures and Tables

**Figure 1 vaccines-11-00559-f001:**
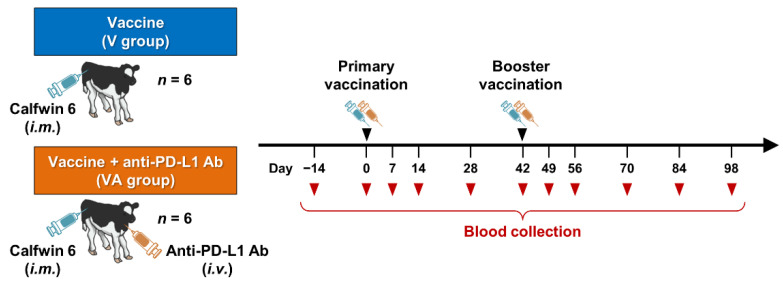
Outline of a pilot study of the combined vaccination with PD-L1 blockade. Twelve calves were administrated intramuscularly with “KYOTOBIKEN” Calfwin 6 Combo Live Vaccine at days 0 and 42. Six of the animals were also intravenously administered 2 mg/kg of anti-PD-L1 Ab just after the vaccinations at days 0 and 42. Peripheral blood samples were collected from all calves at least once every two weeks.

**Figure 2 vaccines-11-00559-f002:**
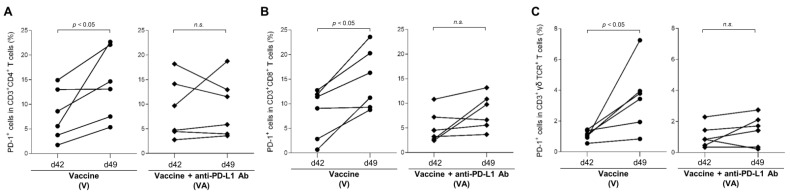
Kinetics of PD-1 expression in T cells before and after vaccination. (**A**–**C**) Proportions of PD-1^+^ cells in CD3^+^CD4^+^ (**A**), CD3^+^CD8^+^ (**B**), and CD3^+^γδTCR^+^ T cell populations (**C**) before (at day 42) and after booster vaccination (at day 49). PD-1 expression in each subpopulation was analyzed by flow cytometry using PBMCs isolated from calves administered Calfwin 6 only (circle, *n* = 6) or Calfwin 6 and anti-PD-L1 Ab (diamond, *n* = 6). Significant differences between each pair of groups were determined using the Mann–Whitney U test. n.s.—not significant. Gating strategies and representative dot plots from the flow cytometric analysis are shown in Appendix A.

**Figure 3 vaccines-11-00559-f003:**
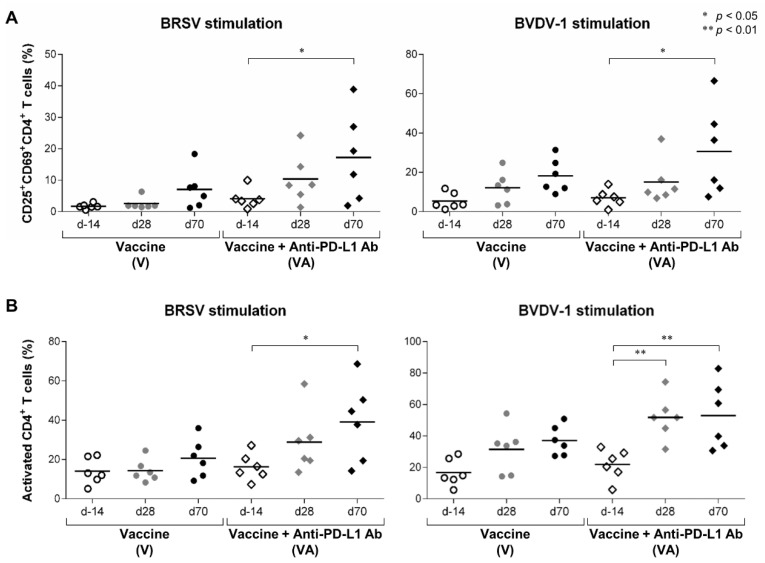
Activation of CD4^+^ T-cell responses to BRSV and BVDV-1 antigens. (**A**,**B**) Proportions of CD25^+^CD69^+^ cells (**A**) and AIM^+^ cells (CD25^+^CD69^+^, CD25^+^CD69^−^, and CD25^−^CD69^+^ cells) (**B**) in the CD3^+^CD4^+^ T-cell fraction of PBMCs stimulated with BRSV (left) or BVDV-1 antigen (right). PBMCs isolated from calves that had been administered Calfwin 6 only (circle, *n* = 6) or Calfwin 6 and anti-PD-L1 Ab (diamond, *n* = 6) at day 14 (white), day 28 (gray), and day 70 (black) were cultured in the presence of BRSV or BVDV-1 antigen for seven days. The expression of CD25 and CD69 in each subpopulation was analyzed by flow cytometry. Gating strategies and representative dot plots for the flow cytometric analysis are shown in Appendix A. Significant differences between groups were determined using Tukey’s test. * *p* < 0.05. ** *p* < 0.01.

**Figure 4 vaccines-11-00559-f004:**
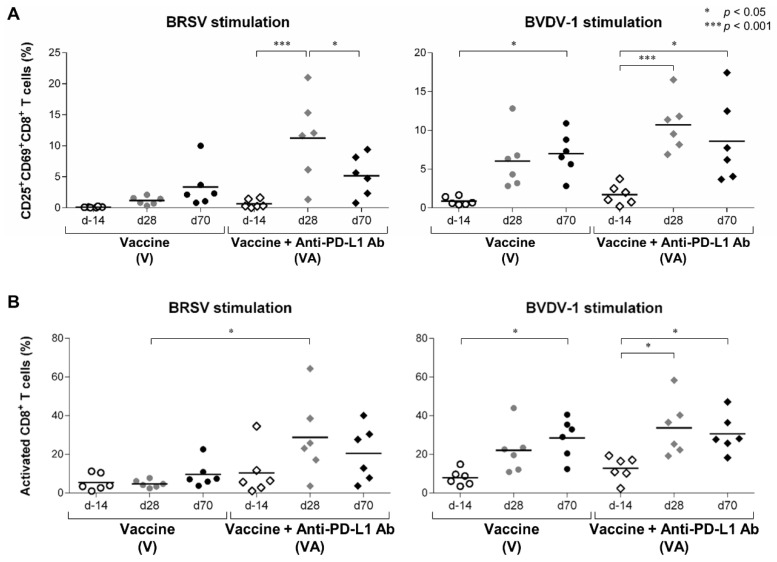
Activation of CD8^+^ T-cell responses following vaccination and PD-L1 blockade. (**A**,**B**) Proportions of CD25^+^CD69^+^ cells (**A**) and AIM^+^ cells (CD25^+^CD69^+^, CD25^+^CD69^−^, and CD25^−^CD69^+^ cells) (**B**) in the CD3^+^CD8^+^ T-cell fraction of PBMCs stimulated with BRSV (left) or BVDV-1 antigen (right). PBMCs isolated from calves that had been administered Calfwin 6 only (circle, *n* = 6) or Calfwin 6 and anti-PD-L1 Ab (diamond, *n* = 6) on day 14 (white), day 28 (gray), and day 70 (black) were cultured in the presence of BRSV or BVDV-1 antigen for seven days. The expression of CD25 and CD69 in each subpopulation was analyzed by flow cytometry. Gating strategies and representative dot plots for flow cytometric analysis are shown in Appendix A. Significant differences among groups were determined using Tukey’s test. * *p* < 0.05. *** *p* < 0.001.

**Figure 5 vaccines-11-00559-f005:**
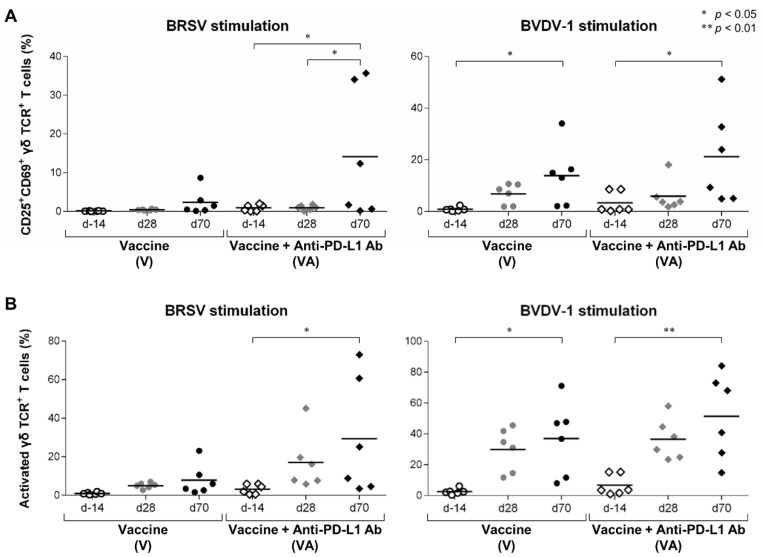
Activation of γδ T-cell responses following vaccination and PD-L1 blockade. (**A**,**B**) Proportions of CD25^+^CD69^+^ cells (**A**) and AIM^+^ cells (CD25^+^CD69^+^, CD25^+^CD69^−^, and CD25^−^CD69^+^ cells) (**B**) in the CD3^+^γδTCR^+^ T cells fraction of PBMCs stimulated with BRSV (left) or BVDV-1 antigen (right). PBMCs isolated from calves that had been administered Calfwin 6 only (circle, *n* = 6) or Calfwin 6 and anti-PD-L1 Ab (diamond, *n* = 6) on day 14 (white), day 28 (gray), and day 70 (black) were cultured in the presence of BRSV or BVDV-1 antigen for seven days. The expression of CD25 and CD69 in each subpopulation was analyzed by flow cytometry. Gating strategies and representative dot plots for flow cytometric analysis are shown in Appendix A. Significant differences among groups were determined using Tukey’s test. * *p* < 0.05. ** *p* < 0.01.

**Figure 6 vaccines-11-00559-f006:**
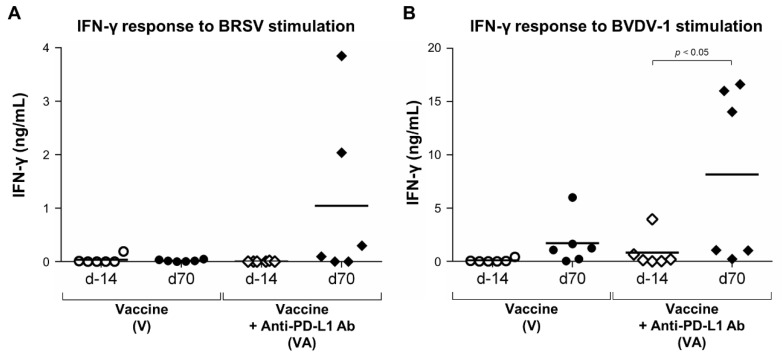
Activation of IFN-γ responses following vaccination and PD-L1 blockade. (**A**,**B**) IFN-γ responses to BRSV (**A**) or BVDV-1 antigen (**B**) before vaccination (on day 14) or after boost vaccination (at day 70). PBMCs isolated from calves that had been administered Calfwin 6 only (left, *n* = 6) or Calfwin 6 and anti-PD-L1 Ab (right, *n* = 6) on days 14 and 70 were cultured in the presence of BRSV or BVDV-1 antigen for seven days. IFN-γ production from PBMCs isolated from each animal was measured by ELISA in duplicate. Significant differences between groups were determined using the Wilcoxon signed-rank sum test.

## Data Availability

The data presented in this study are available on reasonable request from the corresponding author.

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
