# Peer review of "Enhancement of Vaccine-Induced T-Cell Responses by PD-L1 Blockade in Calves"

_vaccines, 2023, doi:10.3390/vaccines11030559_

Round 1

Reviewer 1 Report

Dear Dr. Konnai.

The results obtained are consistent with the experimental design of the study. Additionally, there is a potential translation to the clinical setting of veterinary medicine. However, it would be advisable to consider in the manuscript other in vitro studies that more broadly evaluate the effect of vaccine and of the anti-bovine blocking antibody PD-L1 on the T-cell depletion profile, and also what would be the design of a pilot study in calves. I suggest that these two topics at least be described or mentioned in the manuscript.

Best regards

Author Response

The results obtained are consistent with the experimental design of the study. Additionally, there is a potential translation to the clinical setting of veterinary medicine. However, it would be advisable to consider in the manuscript other in vitro studies that more broadly evaluate the effect of vaccine and of the anti-bovine blocking antibody PD-L1 on the T-cell depletion profile, and also what would be the design of a pilot study in calves. I suggest that these two topics at least be described or mentioned in the manuscript.

Response: Thank you very much for your encouraging comment. We believe that our findings can contribute to the generation of new insights in the immune response to vaccines and the development of improved vaccine programs in calves.

Additional studies to evaluate the blockade effect under T-cell depletion would be worth trying, although not essential. PBMCs collected during this vaccine trial have been cryopreserved. Depending on the storage status of the frozen PBMCs, it may be possible to try this experiment if it would be possible to hold off on submitting a revised manuscript for another month.

We have also provided an overview of the pilot study as Figure 1 to illustrate the design of the study precisely (lines 84 and 158–162 in the marked-up article).

Other revisions:

We propose to change the layout of the text as shown (lines 215–290 in the marked-up article). We also made Figure 2 more compact and displayed the plots side by side. As figures and references are added and modified, their numbers are corrected.

Reviewer 2 Report

The manuscript submitted by Okagawa et al entitled "Enhancement of vaccine-induced T-cell responses by PD-L1 blockade in calves" it's a masterpiece paper. The manuscript is very well-written, reporting a nice set of results that are also very relevant to the field. Briefly, the authors showed that the blockade of the PD-1/PD-L1 interaction enhances T-cell responses induced by vaccination in cattle, indicating the potential utility of anti-PD-L1 Ab in improving the efficacy of current vaccination programs. I would like to congratulate the authors for the conducted research and I would like to ask to correct the following aspects before the resubmition of the work:

1 - Line 93: please clarify if the inoculation of Ab anti-PD-L1 was before, at the same or before vaccination;

2- Figure 1: explain the choosen days (d42 and d49)

3- Figure 291: please add also the info that the PD-1/PD-L1 axis is a pivot in feline tumors (https://doi.org/10.3390/cancers12061386)

Author Response

The manuscript submitted by Okagawa et al entitled "Enhancement of vaccine-induced T-cell responses by PD-L1 blockade in calves" it's a masterpiece paper. The manuscript is very well-written, reporting a nice set of results that are also very relevant to the field. Briefly, the authors showed that the blockade of the PD-1/PD-L1 interaction enhances T-cell responses induced by vaccination in cattle, indicating the potential utility of anti-PD-L1 Ab in improving the efficacy of current vaccination programs. I would like to congratulate the authors for the conducted research and I would like to ask to correct the following aspects before the resubmition of the work:

Response: Thank you very much for your encouraging comment. We believe that our findings can contribute to the generation of new insights in the immune response to vaccines and the development of improved vaccine programs in calves.

1 - Line 93: please clarify if the inoculation of Ab anti-PD-L1 was before, at the same or before vaccination;

Response: Anti-PD-L1 Ab was inoculated just after the primary and booster vaccinations on the same days (lines 94 in the marked-up article).

2- Figure 1: explain the choosen days (d42 and d49)

Response: At day 42, the animals were received the booster vaccination. Peripheral blood samples were collected before the inoculations. Thus, the dataset at days 42 and 49 was chosen to evaluate the response at the day of the booster vaccination (before the booster) and 7 days after the booster (lines 169–172 in the marked-up article).

3- Figure 291: please add also the info that the PD-1/PD-L1 axis is a pivot in feline tumors (https://doi.org/10.3390/cancers12061386)

Response: Two references of the PD-1/PD-L1 axis in feline tumors were cited in the revised manuscript (ref no. 50 and 51, lines 672–677 in the marked-up article).

Other revisions:

We propose to change the layout of the text as shown (lines 215–290 in the marked-up article). We also made Figure 2 more compact and displayed the plots side by side. As figures and references are added and modified, their numbers are corrected.
